# A Low-Latency Approach for RFF Identification in Open-Set Scenarios

**Bo Zhang** [1,2], **Tao Zhang** [1,2,*], **Yuanyuan Ma** [1,2], **Zesheng Xi** [1,2], **Chuan He** [1,2], **Yunfan Wang** [1,2] and **Zhuo Lv** [3]

1   State Grid Smart Grid Research Institute Co., Ltd., Nanjing 210003, China; zhangbo@geiri.sgcc.com.cn (B.Z.); mayuanyuan@geiri.sgcc.com.cn (Y.M.); 4317045xi@163.com (Z.X.); hechuan@geiri.sgcc.com.cn (C.H.); wangyunfan@geiri.sgcc.com.cn (Y.W.)
2   State Grid Key Laboratory of Information & Network, Nanjing 210003, China
3   State Grid Henan Electric Power Research Institute, Zhengzhou 450052, China; lvzhuo@geiri.sgcc.com.cn
*   Correspondence: zhangtao@geiri.sgcc.com.cn

**Abstract:** Radio frequency fingerprint (RFF) identification represents a promising technique for lightweight device authentication. However, current research on RFF primarily focuses on the close-set recognition assumption. Moreover, the high computational complexity and excessive latency during the identification stage represent an intolerable burden for Internet of Things (IoT) devices. In this paper, we propose a deep-learning-based RFF identification framework in relation to open-set scenarios. Specifically, we leverage a simulated training scheme, in which we strategically designate certain devices as simulated unknowns. This allows us to fine-tune our extractor to better handle open-set recognition. Additionally, we construct an exemplar set that only contains representative RFF features to further reduce time consumption in the identification stage. The experiments are carried out on a hardware platform involving LoRa devices and using a USRP N210 software-defined radio receiver. The results show that the proposed framework can achieve 90.23% accuracy for rogue device detection and 93.85% accuracy for legitimate device classification. Furthermore, it is observed that using an exemplar set consisting of half the total data size can reduce the time overhead by 58% compared to using the entire dataset.

**Keywords:** IoT; LoRa; radio frequency fingerprint; deep learning; open-set

## 1. Introduction

The increasing use of wireless devices has raised security concerns regarding device authentication in Internet of Things (IoT) networks, which is essential for allowing legitimate devices to access the network and preventing rogue devices doing so. Traditional identification schemes, such as MAC addresses and cryptographic keys [1], are susceptible to tampering and cracking, respectively [2,3], thereby allowing attackers to disguise themselves as legitimate devices and to access private data. Furthermore, key-based identification suffers from excessive latency with an increasing number of devices due to heavy computation in key management procedures [4–6]. In contrast, use of a radio frequency fingerprint (RFF) is an efficient alternative tool for wireless security that generates a unique fingerprint for each wireless device by leveraging imperfections. The RFF is difficult to modify or tamper with and requires no extra operation from the transmitter [7], making it desirable for power-constrained and low-cost LoRa devices.

However, recent studies on RFF identification have mainly focused on the close-set recognition hypothesis, where the system assumes prior knowledge of all possible wireless communication devices. During testing, there may be unknown classes that the classifier has seldom encountered before [8]. Therefore, in reality, it is difficult to obtain knowledge of all devices, making RFF recognition an open-set problem. If a close-set recognition scheme is utilized in such cases, the system may erroneously accept rogue devices by incorrectly assigning them to one of the known classes due to the limited scope of the deep learning

network [9]. This issue can lead to false negative identifications or incorrect classifications, which can have severe consequences, particularly in security-critical applications. Urgent attention and further research are needed to develop effective strategies to address this issue. Shen et al. [10] introduced a short-time Fourier transform (STFT)-based spectrogram coupled with a CNN for RFF. Their approach exhibited superior identification performance compared to I/Q-based and fast-Fourier-transform-based methods. Furthermore, they investigated the impact of the carrier frequency offset (CFO) on identification performance and proposed the use of an estimated CFO to adjust the identification results, mitigating potential performance degradation. In addressing the challenge of online device authentication scenarios, ref. [11] presented a novel approach utilizing a deep metric learning framework for RFF identification. Additionally, to enhance the robustness of deep learning models towards channel variations, ref. [11] introduced a method leveraging channel-independent spectrograms for feature representation. Ref. [12] introduced "Hawk", an anomaly-based intrusion detection system employing federated learning against emerging new and unknown attacks towards LoRa devices. Leveraging the distinct features of devices, such as the CFO, Hawk achieves effective anomaly detection and exhibits robustness against emerging threats.

Based on the above discussion, this paper proposes a DL-based RFF identification framework that addresses the open-set recognition problem. The framework includes a training method that designates certain devices as unknown classes, allowing for fine-tuning of the feature extractor to better handle open-set recognition. Additionally, an exemplar set is created using partial features from known and simulated unknown classes. These carefully selected features can accelerate the identification process without sacrificing accuracy. The proposed framework's performance is verified through RFF recognition experiments with varying levels of openness. We conducted a comparison between our proposed method and traditional threshold detection as well as the OpenMax [13] method for open-set identification with regard to rogue device detection. Our contributions are summarized as follows:

- We propose a scalable RFF identification framework based on an additional simulated training stage towards open-set. This enhances the learned representation to preserve useful information for separating rogue from legitimate devices, as well as discriminating among legitimate devices.
- We keep the representative features from the training data to construct a feature exemplar set that efficiently characterizes the RFF patterns of legitimate and rogue devices. This could help to reduce the feature space and computational complexity during the testing process.
- We verify the effectiveness of the proposed approach by utilizing LoRa devices and a USRP N210 software-defined radio (SDR) platform. The experimental results show that the accuracy of rogue device identification and device classification is higher than when using the thresholds directly and other open-set algorithms, such as OpenMax. Moreover, the exemplar set achieves over 90% accuracy even with half of the total features, with a shorter recognition time.

The rest of the paper is organized as follows: Section 2 introduces related work. Section 3 provides a system overview. Sections 4 and 5 introduce LoRa signal processing and RFF extractor training, respectively. Section 6 presents the simulated training and the procedure for rogue device detection and legitimate device classification. Section 7 provides extensive experimental results to demonstrate the system performance. Section 8 concludes the paper.

## 2. Related Work

The rapid development of IoT has increased the need for a reliable and low-latency authentication scheme to ensure that RFF can be widely deployed. However, as wireless communication research advances, meeting the demand for low latency in practical deployment scenarios has become increasingly challenging. Das [14] pointed out that wireless

communication's reliability and latency hinder industrial wireless control systems' progress. To address this issue, Weiner et al. [15] proposed a wireless system architecture based on semifixed resource allocation and low-rate coding. In addition, Zheng et al. [16] proposed a lightweight RFF recognition network based on a compact ResNet architecture to improve RFF recognition speed and reduce computational complexity. Another study proposed an embedded RFF recognition method based on a lightweight network architecture [17]. The proposed method uses a combination of convolutional and recurrent neural networks to extract RFF features and classify them. However, these works do not provide a detailed evaluation of the proposed method's limitations or its performance in scenarios with high levels of noise or interference, which are common in many real-world applications.

Most of the existing RFF identification works focus on deep learning (DL) methods [18–20]. However, the majority of DL-based RFF research has been conducted based on the close-set hypothesis, which assumes that the prior probability of all wireless communication devices is known. In reality, there are many unknown devices in most scenarios, and many previous DL-based RFF identification schemes lack scalability in the open-set environment [10]. This is because previous methods usually rely on the softmax layer for classification, and once training is completed, the number of neurons in this layer cannot be changed, turning RFF identification into a close-set problem [21]. Furthermore, this design raises concerns that the DL model needs to be retrained whenever unknown devices are present in the training set, leading to greater time-consumption. Additionally, rogue devices are not readily predicted, and their data is often not available for training. As a result, during identification, they will be classified as legitimate devices with the most similar characteristics to those in the training categories [22], and this is not acceptable.

In order to overcome this issue, in 2020, Geng et al. [8] conducted a survey on recent advances in open-set recognition. Their findings showed that recent research mainly focused on computer vision and pattern recognition, with limited studies on RFF open-set recognition. However, in recent years, scholars have carried out research on RFF open-set recognition. In 2021, Fang et al. [4] proposed an end-to-end RFF open-set recognition method based on hypersphere representation. Furthermore, the OpenMax model [13] based on meta-recognition theory [23] provides a solution to the DL network towards an open-set problem without requiring extra retraining.

Compared with traditional DL methods, our approach emphasizes the distinguishability of feature distances in the feature space by applying metric learning, which means that devices of the same class are brought closer together, while devices of different classes are pushed farther apart. Furthermore, we introduce a simulated training process to create an exemplar set, effectively representing RFF features and reducing the time overhead without significantly influencing the identification accuracy. This allows us to refine our feature extractor and enhance its capacity for open-set identification.

## 3. System Overview

The proposed framework utilizes a DL-based RFF identification framework. It consists of three stages, namely, training, simulated training, and identification, as shown in Figure 1.

**Training:** During the training stage, we utilize a DL-based extractor model to extract distinctive RFF features from training packets collected from various legitimate devices. These packets undergo preprocessing steps, such as synchronization and CFO estimation and compensation, and are labeled accordingly. The RFF extractor is trained based on these preprocessed signals to extract device-specific RFF features. The training stage is only performed once, whereas after the extractor is trained, we construct an exemplar set by selecting representative RFF and CFO features from the extractor. This exemplar set is then used to train an RFF classifier.

**Simulated Training:** During the simulated training stage, we utilize the exemplar set to train an RFF classifier. After the training process, we split the exemplar set into two parts, one part for legitimate devices and the other part for simulating rogue devices. Some

data from the legitimate devices part are used to fit the K-nearest neighbors (KNN) model. Then, the KNN model predicts the labels of the remaining data from the exemplar set. We use the remaining data and labels given by the KNN model to train the logistic regression (LR) model. During the device identification stage, the KNN model is mainly responsible for the legitimate device classification function, while the LR model is responsible for the rogue device detection function.

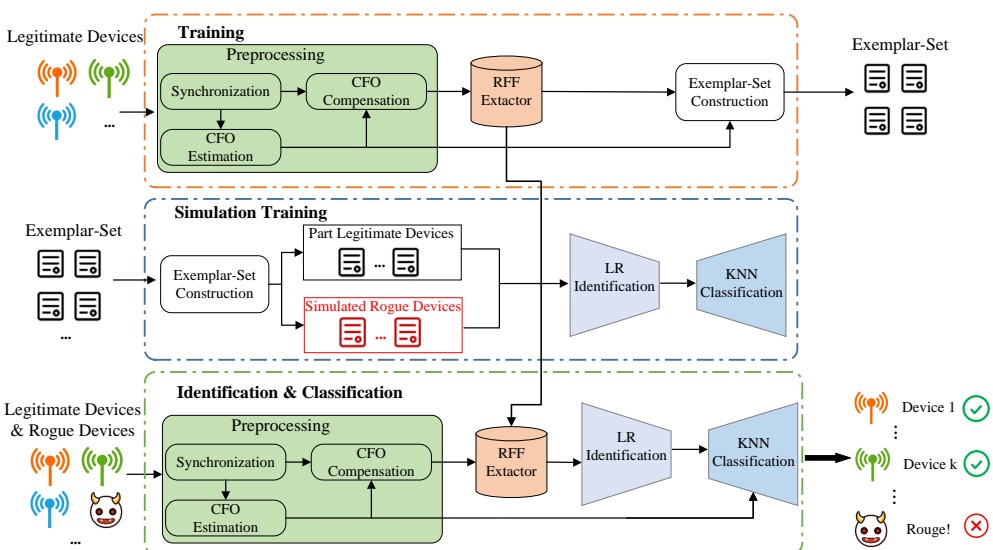

**Figure 1.** System framework of the proposed RFF identification system.

**Identification and Classification:** The two main processes in the identification stage are rogue device detection and legitimate device classification. First, preprocessing is performed on the received signal in the same way as during the training phase. The RFF features are then extracted from the preprocessed signal using the RFF extractor. The legitimacy of the packets can be determined by applying the LR model. The legitimate device is then classified using the KNN model if it is. If not, the packet is identified as coming from a rogue device, and the access will be prevented.

## 4. LoRa Signal Preprocessing

In this section, we introduce the preprocessing method of the signal and then illustrate the process of construction of the exemplar set.

### 4.1. LoRa Signal

LoRa uses chirp spread spectrum (CSS) technology for modulation, where chirps are used to communicate. A basic LoRa symbol can be written as

$$x(t) = Ae^{j2\pi(-\frac{b\omega}{2} + \frac{b\omega}{2T}t)t} \quad (0 \leq t \leq T), \tag{1}$$

where $A$ and $b\omega$ denote the amplitude and bandwidth, respectively, and $T$ is the LoRa symbol duration. The preamble, which consists of eight repeating $x(t)$, is present at the beginning of every LoRa packet and is identical for all device types.

### 4.2. Signal Acquisition

The transmitted baseband signal $x(t)$ undergoes signal modulation and up-conversion via hardware components, such as an oscillator and power amplifier. These components introduce their specific impairments, and the overall effect of device $i$ is denoted as $\mathcal{F}_i(\cdot)$. The signal is then captured by the receiver. The received baseband signal $r_i(t)$ of device $i$ is given by

$$r_i(t) = h(t) * \mathcal{F}_i(x(t)) + v(t), \tag{2}$$

where $h(t)$ is the time-varying channel impulse response, $v(t)$ is the additive white Gaussian noise, and $*$ denotes the convolution operation. When the receiver captures a packet from device $i$, it extracts the preamble part $r_i(t)$ and digitizes it, denoted as $r_i(n)$.

*4.3. Preprocessing*

To make sure the received signal satisfies the fundamental requirements of the RFF identification procedure, preprocessing is required. This covers normalization, preamble extraction, synchronization, and CFO compensation. These algorithms are explained in more detail in [24].

**Synchronization:** In this step, the packet's starting point is identified. Precise synchronization is essential because a segment of channel noise is introduced by imprecise synchronization, which degrades RFF identification performance.

**Preamble Extraction:** To avoid the deep learning model capturing identity-related details, such as the MAC address, when leveraging the entire packet for RFF, we selectively use only the preamble part.

**CFO Compensation:** CFO compensation is required to ensure system stability in RFF identification. Crystal oscillators are very sensitive to temperature variations. System performance can be severely impaired by oscillator frequency drift [25].

**Normalization:** Normalization keeps the non-device-specific received power from being learned by the system. By dividing its root mean square (RMS), the preamble portion is normalized.

## 5. Rff Extractor Training

The RFF extractor, which is an essential module in the proposed RFF system, should be capable of generalizing well towards open-set scenarios for the extraction of RFF features of unknown devices.

*5.1. Deep Learning Model Architecture*

To efficiently extract the RFF features from the LoRa signal, we treat it as a time sequence and employ a convolutional neural network (CNN) as our classification model. Our goal is to design a deep neural network capable of fully extracting the RFF features from the preprocessed signal. The network architecture is inspired by the ResNet architecture but is optimized to be lightweight and suitable for the characteristics of LoRa signal data.

The architecture of our RFF extractor, illustrated in Figure 2, is based on a residual neural network (ResNet) structure with cross-layer connections. This architecture consists of nine convolution layers, one average pooling layer, and one dense layer with 512 neurons. The RFF extractor takes a 1024-dimensional vector uniformly sampled from the preprocessed LoRa signal as input. The first convolution layer comprises 32 filters with a size of $7 \times 1$ and a stride of 2. Subsequent convolution layers (the second to fifth) consist of 64 filters with a size of $3 \times 1$. The sixth to ninth convolution layers employ 32 filters with the same size as the previous layers. To capture non-linear relationships, all the convolutional layers are activated using rectified linear units (ReLU). Additionally, the strategic use of padding in convolutional layers preserves the spatial information of the input signal, preventing the loss of crucial features. The output of the last convolutional layer undergoes average pooling before being fed into the dense layer. The L2 normalized model produces a 512-dimensional vector, representing the RFF features extracted from the received packets.

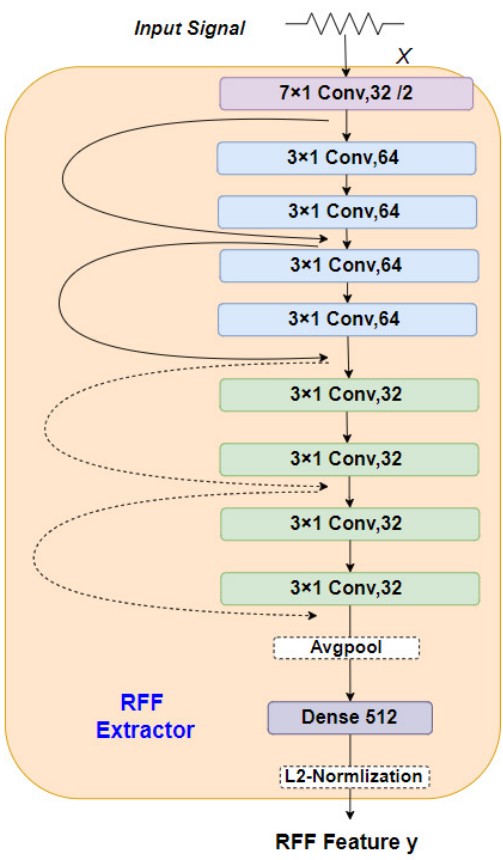

**Figure 2.** RFF feature extractor based on ResNet.

### *5.2. Deep Metric Learning*

Deep metric learning is a technique used to train neural networks to learn embeddings. It can represent input data in a space where samples with the same labels are close to each other and samples with different labels are far apart. Unlike traditional methods that rely solely on capturing patterns, deep metric learning enables learning of a more discriminative and semantically meaningful representation of the RFF data. In this section, the RFF extractor is trained using deep metric learning and triplet loss as the loss function.

The triplet loss, initially developed for face recognition [26], has gained popularity in RFF recognition due to its ability to learn meaningful embeddings and enforce desirable distance relationships between samples. It compares the distance between the embeddings of an anchor sample, a positive sample (same label as anchor), and a negative sample (different label than anchor). The triplet network uses a shared weight network to minimize the distance between the anchor and positive samples and to maximize the distance between the anchor and negative samples. Figure 3 illustrates a triplet consisting of an anchor ($\mathbf{x}^{an}$), a positive ($\mathbf{x}^{+}$), and a negative ($\mathbf{x}^{-}$) sample, where the anchor and positive samples are from the same device and the negative sample is from a different device. Mathematically, the triplet loss can be expressed as:

$$\mathcal{L}_{Triplet} = \max\left[||\mathbf{x}^{+} - \mathbf{x}^{an}||_2 - ||\mathbf{x}^{-} - \mathbf{x}^{an}||_2 + \xi, 0\right], \tag{3}$$

where $|| \cdot ||_2$ is $\ell_2$ norm, which denotes the Euclidean distance between two vectors. $\xi$ is a parameter in the loss function, which denotes the margin between the positive and negative pairs. By leveraging deep metric learning based on triplet loss, as shown in Figure 3, the feature embedding distances between the anchor and the negative sample are larger, while the distances between the anchor and the positive sample are smaller.

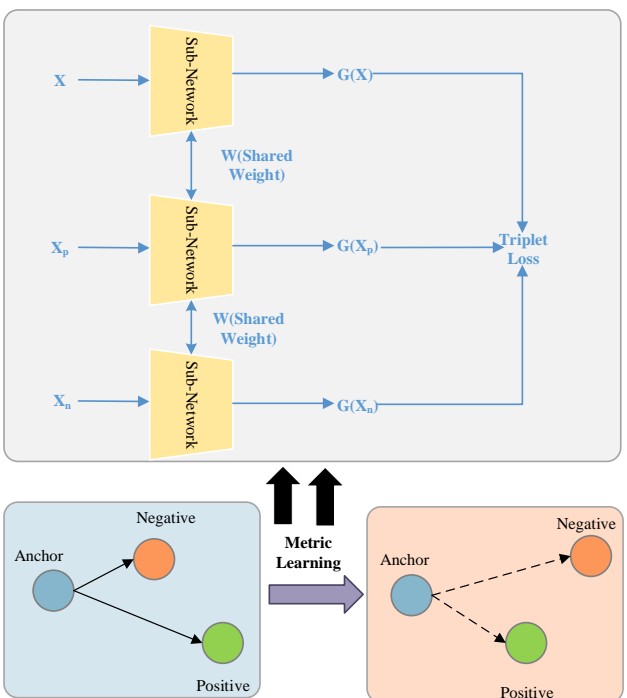

**Figure 3.** Metric-learning-based triplet loss.

*5.3. Exemplar Set Construction*

We create an exemplar set by choosing exemplars from the RFF extractor's outputs after the training phase. We built the exemplar set using a fusion input of RFF signals and the CFO to improve the efficacy of RFF verification. The observation that employing the CFO and RFF signals separately does not produce satisfactory results served as the driving force behind this decision.

The exemplar set construction algorithm is used to select a subset of exemplars from the outputs of the RFF extractor. The RFF extractor is trained on the dataset $X$ and the CFO dataset $C$ of the device $y$. Algorithm 1 takes as input the target number of exemplars $M$ for each class, and the current RFF extractor $\varphi$, which maps the input signal $x$ to a $d$-dimensional feature vector in $R^d$.

For each device $y$, the algorithm first calculates the mean feature vector $\mu$ of all the samples in $X$. Then, for each class, the algorithm selects the $M$ closest samples to $\mu$ using the Euclidean distance as the similarity metric. For each of these $M$ samples, the algorithm records its RFF feature vector $\varphi(s_k)$ and its CFO value $c_k$. The resulting set of exemplars for device $y$ is denoted $S_y$ and consists of the $M$-selected samples along with their corresponding RFF feature vectors and CFO values. In addition to constructing the exemplar set, the algorithm also records the minimum and maximum CFO values of each legitimate device in the training dataset. These values are stored in the variables $\text{CFO}_{\min_y}$ and $\text{CFO}_{\max_y}$, respectively.

The final output of the algorithm is the set of exemplars $S$, the minimum CFO values $\text{CFO}_{\min}$ dataset, and the maximum CFO values $\text{CFO}_{\max}$ dataset for each device. The exemplar set is smaller than the original dataset since only a subset of samples is selected, and the size can be controlled by changing the value of $M$. The RFF extractor can then be used to extract the 512-dimensional RFF features from the input signal using the exemplar set.

---

**Algorithm 1** Exemplar Set Construction

---

**Require:** RFF data $X_y = \{x_1, \ldots, x_n\}$ of device $y$
**Require:** CFO data $C_y = \{cfo_1, \ldots, cfo_n\}$ of device $y$
**Require:** $M$ exemplars for each class
**Require:** Current RFF extractor $\varphi : x \to \mathbb{R}^d$
  **for** $y = 1, \ldots, Y$ **do**
    $\mu \leftarrow \frac{1}{n} \sum_{x \in X_y} \varphi(x)$
    **for** $k = 1, \ldots, M$ **do**
      $s_k \leftarrow \underset{x \in X_y}{\mathrm{argmin}} \|\mu - \varphi(x)\|$
      $c_k \leftarrow cfo_{X.\,\mathrm{index}(s_k)}$
      $s_k \leftarrow (\varphi(s_k), c_k)$
    **end for**
    $S_y \leftarrow \{s_1, \ldots, s_M\}$
    $\mathrm{CFO}_{\min_y} \leftarrow \underset{cfo \in C_y}{\min} (cfo)$
    $\mathrm{CFO}_{\max_y} \leftarrow \underset{cfo \in C_y}{\max} (cfo)$
  **end for**
  $S \leftarrow \{S_1, \ldots, S_Y\}$
  $\mathrm{CFO}_{\min} \leftarrow \{\mathrm{CFO}_{\min_1}, \ldots, \mathrm{CFO}_{\min_Y}\}$
  $\mathrm{CFO}_{\max} \leftarrow \{\mathrm{CFO}_{\max_1}, \ldots, \mathrm{CFO}_{\max_Y}\}$
**Ensure:** $S$, $\mathrm{CFO}_{\min}$, and $\mathrm{CFO}_{\max}$

---

## 6. Device Identification and Verification

After the training of the RFF feature extractor, we focus on training a robust classifier using the exemplar set constructed in Section 5.3, and provide a detailed description of the simulated training and device identification processes.

The features after the extractor are designed to capture two important characteristics: the CFO and distance information. The CFO is caused by inherent hardware defects in devices, and the frequency offset of different devices can vary significantly. As for the distance information, the RFF distances between the samples from legitimate devices and rogue devices are quite distinct in the feature space. Therefore, we design two features, $\alpha$ and $\beta$, which denote the feature distance and the CFO, respectively.

$$\mathbf{z} = [\alpha, \beta], \tag{4}$$

$$\alpha = \mathbf{D}_{min_k}, \tag{5}$$

$$\beta = (\mathrm{CFO} - \frac{\mathrm{CFO}_{min} + \mathrm{CFO}_{max}}{2}) \cdot \mathrm{sgn}(\mathrm{CFO}) \tag{6}$$

The $\mathbf{D}_{min_k}$ in (5) is the minimum distance between the test samples and their $k$ nearest neighbors in the KNN model. The $\mathrm{CFO}_{min}$ and $\mathrm{CFO}_{max}$ in (6) are the minimum and maximum CFO values of the training samples with the same label as the predicted device, respectively, while $\mathrm{sgn}(\cdot)$ represents the sign function, which indicates the sign of CFO (1, 0, or $-1$). This formulation captures the directional deviation of $\beta$ concerning CFO and the midpoint.

After that, the exemplar set undergoes division into two distinct components: legitimate device verification and simulated rogue device identification. The device verification stage is to determine the specific labels of the devices from the signal, which outputs a legitimate label, which is implemented by the KNN. The RFF of the received packets is first extracted by the RFF extractor, and then the features will be selected according to (4). The most frequent label among the K-neighbors is assigned to the device.

Rogue device detection is a process to identify whether a received signal is transmitted from a legitimate device or not. It involves the use of both KNN and LR models. First, the

KNN model calculates the distance between the test sample and the exemplar set data. Then, we use the minimum distance between the test samples and the K-nearest samples to calculate features based on Equations (5) and (6). Next, these features are used as input to the LR model, which determines whether the packet is from a rogue device. The LR model, with output $f(\mathbf{z}) = \frac{1}{1+e^{-\theta \mathbf{z}}}$, distinguishes legitimate from rogue devices. A "True" prediction indicates a legitimate device and then is classified as the output label of the KNN model, a "False" prediction implies a rogue device.

## 7. Experiments

### 7.1. Experimental Settings

The data collection system in Figure 4 consists of 35 target LoRa SX1262 devices for testing and a USRP N210 software-defined radio (SDR) platform as the receiver. The carrier frequency $f_c$ = 433 MHz, bandwidth $B$ = 125 kHz, and $SF = 7$ are the settings for all LoRa devices. The configuration of the receiver is a 2 MHz sampling rate and $f_c$ = 433 MHz.

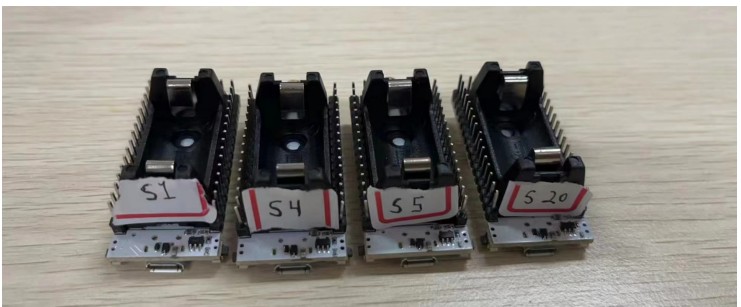

(**a**) Target LoRa devices

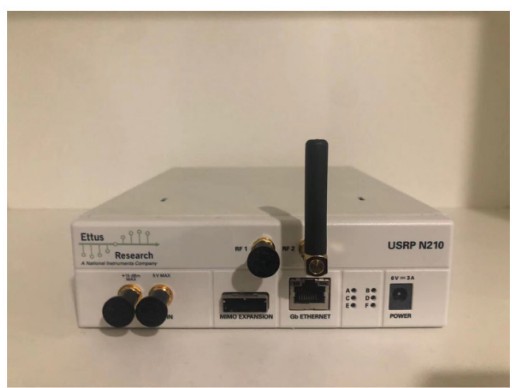

(**b**) USRP

**Figure 4.** Experimental devices. (**a**) Target LoRa devices; (**b**) USRP N210 SDR.

From each LoRa device, we collected 470 packets between the transmitter and the receiver. The 470 packets were divided into two groups: the remaining 70 packets were set aside for testing, and the other 400 training data were chosen at random. All known legitimate device packets, along with supplemented data from rogue device simulations, were used to train the RFF extractor. The rest of the packets, including those from unknown devices, were set aside for identification. The deep learning model was implemented using PyTorch. The training procedure was carried out over 70 epochs under the guidance of the triplet loss function. The learning rate started at 0.002 and decreased every 10 epochs by a factor of 0.98. A 64-batch size was used when applying SGD optimization.

### 7.2. Preliminaries

When it comes to identifying rogue devices, a major challenge arises due to the lack of training data available for them. The accuracy of identifying rogue devices is affected by

the different levels of openness in the task. To evaluate the effectiveness of our proposed method, we use multiple metrics.

(1) *Openness:* The openness [27] is chosen to symbolize the intricacy of open-set identification in various experimental setups, which is denoted as:

$$\mathbf{O}^* = 1 - \sqrt{\frac{2 \times C_{\text{TR}}}{C_{\text{TR}} + C_{\text{TE}}}}, \tag{7}$$

where $C_{\text{TR}}$ and $C_{\text{TE}}$ represent the number of devices in the training samples and testing samples. The larger the value of $\mathbf{O}^*$, the more difficult the problem is.

(2) *Overall Accuracy of Two Tasks:* We calculate three types of accuracy to evaluate the performance of the methods, which include the legitimate classification accuracy (*L*-acc), the rogue detection accuracy (*R*-acc), and the overall accuracy (*O*-acc). *L*-acc represents the correctly classified legitimate devices among all legitimate devices, while *R*-acc represents the correctly detected rogue devices among all devices. *O*-acc is the balance of the *L*-acc and *R*-acc, which evaluates the overall method performance. They are defined as:

$$
\begin{aligned}
L\text{-acc} &= \frac{T_L}{T_L + F_L}, \\
R\text{-acc} &= \frac{T_R}{T_R + F_R}, \\
O\text{-acc} &= \frac{2 \times L\text{-acc} \times R\text{-acc}}{L\text{-acc} + R\text{-acc}} \\
&= \frac{T_L + T_R}{T_L + F_L + T_R + F_R}.
\end{aligned}
\tag{8}
$$

where $T_L$ and $F_L$ correspond to the true correctly classified legitimate device data and the false incorrectly classified legitimate device data, respectively. $T_R$ and $F_R$ refer to the true correctly and false incorrectly classified rogue device data, respectively. The *O*-acc is the balance of the *L*-acc and *R*-acc, which evaluates the overall method performance by applying its harmonic mean.

(3) *Device Identification Metrics*: The task of rogue device detection can be regarded as binary classification. Therefore, precision, recall, specificity, and the F1-score are used to evaluate the performance of the proposed model via the confusion matrix. These performance metrics are defined as follows:

$$\text{Precision} = \frac{TP}{TP + FP}, \tag{9}$$

$$\text{Recall} = \frac{TP}{TP + FN}, \tag{10}$$

$$\text{F1-score} = 2 \times \left( \frac{\text{Precision} \times \text{Recall}}{\text{Precision} + \text{Recall}} \right), \tag{11}$$

$$\text{Specificity} = 1 - \frac{TN}{FP + TN}, \tag{12}$$

where $TP$, $TN$, $FP$, and $FN$ are the true positives, true negatives, false positives, and false negatives, respectively. The detailed performance metrics of the different methods are shown in Table 1.

Here, we compare the different device verification methods used in our study to classify packets received from devices as legitimate or rogue, adopting the same RFF extractor. OpenMax, originally designed for image classification [13], is adapted by training the CNN network and determining the score of a signal from an unknown device using the OpenMax layer. Another approach involves setting a CFO threshold ($\lambda$) to classify packets as rogue if the detected CFO is above $\lambda$. Similarly, a distance threshold ($D$) is employed, categorizing packets as rogue if the minimum distance in the KNN model

exceeds *D*. A combined approach utilizes the CFO and distance thresholds, where a packet is considered rogue if either the distance or the CFO surpasses its threshold. Additionally, logistic regression (LR) is employed, training the model using an exemplar set to discern whether a packet is from a rogue device. Different device configurations are explored to evaluate the LR model's performance under varied conditions, with 18, 20, and 25 devices designated as legitimate and 17, 15, and 10 as rogue, while setting *M* in the exemplar set construction to 200.

The device verification results shown in Table 2 highlight the effectiveness of the strategy of the two parts of the task separately across different openness levels. At a low openness level (0.1548), LR outperforms the other methods, achieving a high legitimate classification accuracy (*L*-acc) of 93.8%, a robust rogue detection accuracy (*R*-acc) of 90.2%, and an impressive overall accuracy (*O*-acc) of 92.5%. In comparison, the other methods, such as Openmax, CFO, distance, and CFO plus distance exhibit lower performance metrics. As the openness level increases to 0.2441, LR maintains its superiority with an 86.5% overall accuracy, surpassing alternative methods such as Openmax, CFO, distance, and CFO plus distance. Even at a higher openness level of 0.2829, LR demonstrates robust performance, achieving an 80.2% overall accuracy. This performance consistency across various levels of openness highlights the validity of the effectiveness of our approach in distinguishing between legitimate and rogue devices in open-set scenarios.

**Table 1.** The performance of device verification.

| Device Verification | | | | |
|---|---|---|---|---|
| **Openness** | **Methods** | **L-acc** | **R-acc** | **O-acc** |
| | LR & KNN | 0.938 | 0.902 | 0.925 |
| | Openmax | 0.925 | 0.849 | 0.889 |
| 0.1548 | CFO | 0.881 | 0.821 | 0.869 |
| | Distance | 0.924 | 0.673 | 0.801 |
| | CFO & Distance | 0.872 | 0.769 | 0.848 |
| | LR & KNN | 0.921 | 0.828 | 0.865 |
| | Openmax | 0.872 | 0.804 | 0.851 |
| 0.2441 | CFO | 0.874 | 0.781 | 0.831 |
| | Distance | 0.911 | 0.626 | 0.807 |
| | CFO & Distance | 0.861 | 0.739 | 0.821 |
| | LR & KNN | 0.916 | 0.753 | 0.802 |
| | Openmax | 0.835 | 0.785 | 0.796 |
| 0.2829 | CFO | 0.747 | 0.711 | 0.747 |
| | Distance | 0.908 | 0.518 | 0.778 |
| | CFO & Distance | 0.723 | 0.647 | 0.723 |

From Figure 5, we can see that our method has better performance than other methods in *L*-acc. When openness is 0.1548, corresponding to the selection of 25 devices as legitimate devices, the *L*-acc of our method reaches 93.85% on 35 devices, which is higher than the 92.57% of OpenMax, and the other threshold methods. As the number of unknown devices gradually increases and the openness increases, the *L*-acc performance of all the methods decreases. Nevertheless, our proposed method can still achieve an accuracy of 90.72% even under high levels of openness, while other methods show a rapid decline in performance. In terms of *R*-acc, our method shows a rapid performance degradation with increase in openness. In contrast, OpenMax performs better in identifying unknown classes, indicating that it is well-suited to open-set recognition problems. *O*-acc measures the combined effect of both *L*-accand *R*-acc. From the perspective of *O*-acc performance, our proposed method performs with 92.14% *O*-acc when the openness is 0.1548, which is better than OpenMax and the other methods. However, as the openness increases, the performance of recognizing unknown classes decreases rapidly, which needs to be improved in future work.

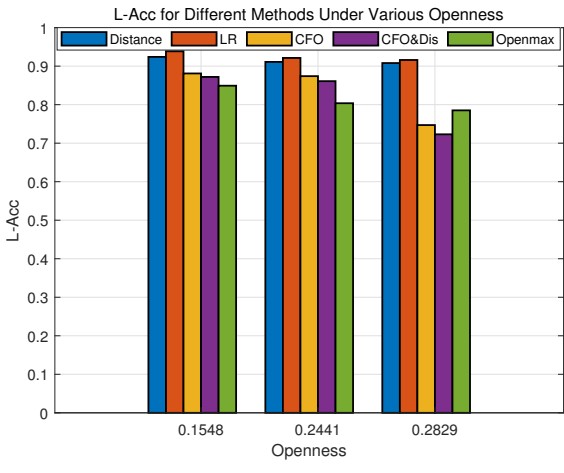

(**a**) L-Accuracy

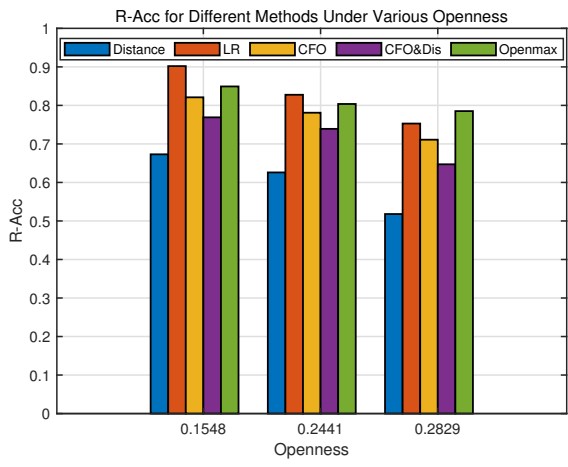

(**b**) R-Accuracy

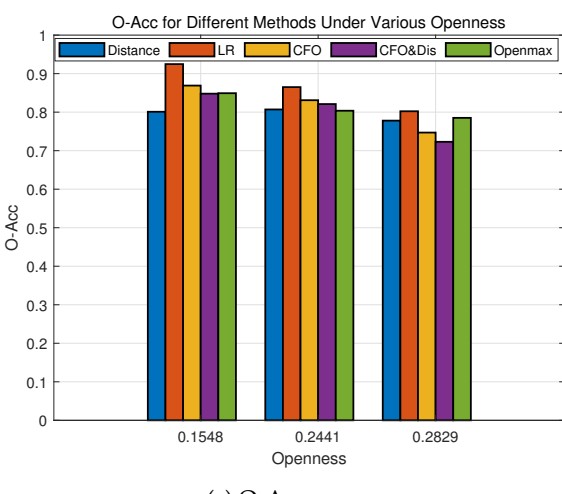

(**c**) O-Accuracy

**Figure 5.** Accuracy performance of different methods.

### 7.3. The Effect of Device Identification

Moreover, we evaluate the performance of our proposed method by constructing the confusion matrix for legitimate device classification and the receiver operating characteristic curve (ROC curve) for rogue device detection using 25 devices for training. The

confusion matrix, depicted in Figure 6, indicates that most samples are accurately classified; however, certain categories are frequently misclassified. Specifically, 12 samples belonging to Category 3 are misclassified as Category 2, among others.

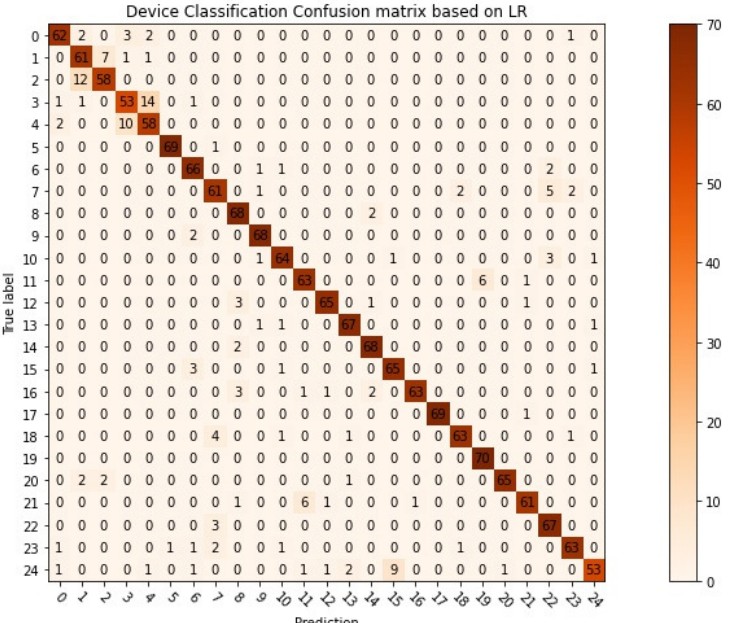

**Figure 6.** Confusion matrix about the legitimate device classification of our method.

As well as the confusion matrix, as Figure 7 illustrates, we use ROC curves to assess the performance of rogue device detection. The ROC curve plots the false positive rate ($FPR$) on the x-axis and the true positive rate ($TPR$) on the y-axis, calculated as follows:

$$FPR = \frac{FP}{N} = \frac{FP}{FP + TN}. \tag{13}$$

$$TPR = \frac{TP}{P} = \frac{TP}{TP + FN}. \tag{14}$$

Figure 7 illustrates the ROC curve of the simulated attack detection and the corresponding area under the curve (AUC) values. With an AUC value of 0.92, our LR approach is a notable performer demonstrating its ability to discriminate between rogue and legitimate devices. This performance is much better than that of the threshold-based detection techniques, such as the CFO and distance, and it outperforms the OpenMax method (AUC = 0.89). Remarkably, the LR curve is consistently dominant, confirming its effectiveness in the open-set task of detecting rogue devices.

Table 2 shows the effect of using different methods on the confusion matrix, with a focus on the F1-score, recall, precision, and specificity. The LR model achieves the highest F1-score of 0.9080, followed by OpenMax with 0.9022, while the CFO and distance methods have lower F1-scores of 0.8732 and 0.6347, respectively. The LR model also has the highest recall and precision, indicating a better ability to correctly identify both rogue and legitimate devices. Additionally, all methods have high specificity, with LR and OpenMax having values close to 1, indicating a low false positive rate. Overall, the table highlights the effectiveness of the LR model for device authentication, with better performance than the other methods.

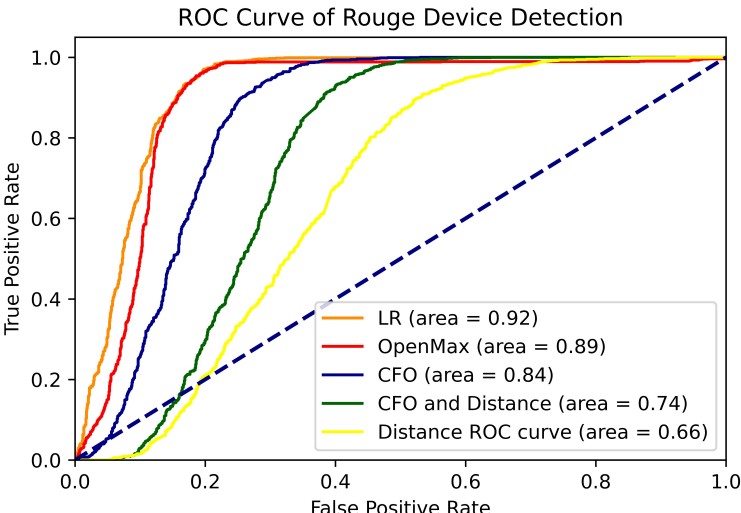

**Figure 7.** ROC curve of rogue device detection under different methods.

**Table 2.** The performance of device identification.

| | Device Identification | | | |
|---|---|---|---|---|
| **Methods** | **Precision** | **F1-Score** | **Recall** | **Specificity** |
| LR | 0.9104 | 0.908 | 0.9081 | 0.9962 |
| Openmax | 0.9049 | 0.9022 | 0.9023 | 0.9959 |
| CFO | 0.8816 | 0.8732 | 0.8794 | 0.9892 |
| Distance | 0.6391 | 0.6347 | 0.6376 | 0.9693 |

*7.4. The Effect of Exemplar Set*

In this section, we present and discuss experiments that demonstrate the effect of the exemplar set size on the accuracy and identification time for our proposed method. Specifically, we construct exemplar sets of different sizes, including $M = 20, 40, 60, 100, 200,$ and $400$; $M = 400$ means the whole of the training data. The exemplar set is selected from all the training data; thus, it has fewer data than the whole samples, resulting in lower time cost.

Figure 8 illustrates the *O*-acc and identification time cost of the different exemplar set sizes. The LR model trained with exemplar set construction achieves an O-accuracy of 92.14% and an identification time cost of 18.21 s when using all the training data ($M = 400$). However, with a smaller exemplar set size of $M = 100$, the time cost is reduced to 6.73 s, while maintaining a slightly lower O-accuracy of 90.82%. The results show its ability to accelerate the training process while maintaining high accuracy.

*7.5. The Effect of SNR*

In this section, we describe the experiments conducted to evaluate the performance of our method under varying SNRs. We first train the model using data with relatively high original SNR under various openness. Subsequently, we evaluate the model's recognition accuracy on multiple test datasets with varying Gaussian white noise levels, ranging from 0 dB to 20 dB. The OpenMax method is employed as a benchmark for comparison.

As illustrated in Figure 9, it is evident that the accuracy of both LR and OpenMax increases with higher SNR settings. Notably, as the degree of openness rises, there is a decline in performance, potentially attributed to insufficient classifier training due to the presence of more unknown samples. In the low-SNR region (0 dB to 5 dB), LR may exhibit suboptimal performance compared to the OpenMax method, exhibiting a 1.83% accuracy gap. However, under higher SNR conditions, LR outperforms OpenMax, which achieves

92.76% accuracy when the setting of the openness is 0.1548 and 20 dB. The observed poorer performance of LR under low-SNR conditions is attributed to noise disrupting the original data distribution, leading to a deviation from linear features in both the input and output data. This phenomenon ultimately results in a decline in performance.

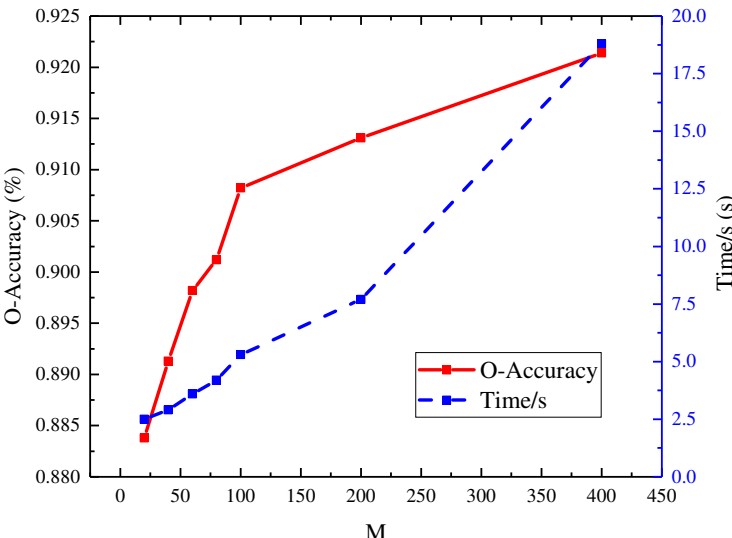

**Figure 8.** Time and O-accuracy of different exemplar sets.

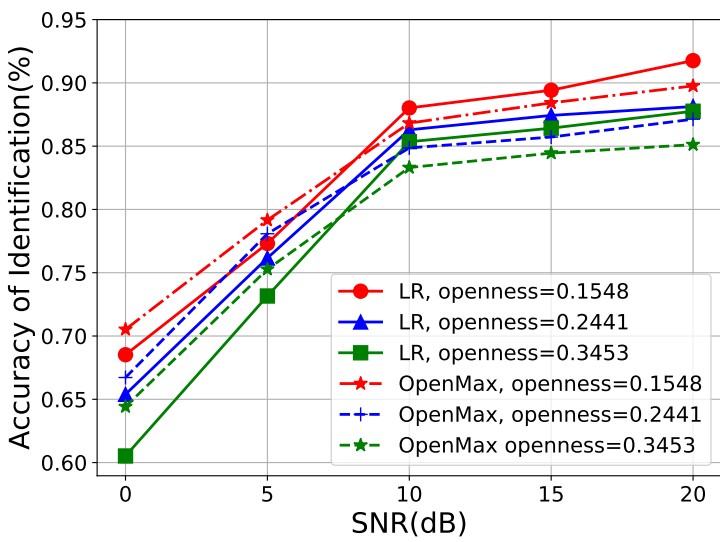

**Figure 9.** The performance under different SNRs.

## 8. Conclusions

In conclusion, we introduced a novel DL-based RFF identification framework leveraging device-intrinsic hardware impairments for robust device authentication. Our approach incorporated a simulated training process to effectively train an RFF extractor with remarkable generalization capabilities. Additionally, the implementation of an exemplar set mitigated the time overhead during the identification stage. For rogue device detection, we employed an LR model, while legitimate device classification was accomplished using the KNN algorithm. The extensive experiments conducted validated the performance of our method in both rogue device detection and legitimate device classification. Notably, our constructed exemplar set demonstrated its efficacy in significantly accelerating the training process with minimal impact on accuracy.

**Author Contributions:** Conceptualization, B.Z. and T.Z.; methodology, T.Z.; software, Y.M.; validation, B.Z., Z.X. and C.H.; formal analysis, B.Z., C.H. and Y.W.; investigation, C.H. and Y.W.; resources, Y.M.; data curation, Z.X. and Z.L.; writing—original draft preparation, B.Z.; writing—review and editing, T.Z.; visualization, Y.M. and Z.L.; supervision, T.Z.; project administration, Z.X.; funding acquisition, T.Z. All authors have read and agreed to the published version of the manuscript.

**Funding:** This research was supported in part by the National Key R&D Program of China (No. 2022YFB3104300).

**Data Availability Statement:** Data is contained within the article.

**Conflicts of Interest:** Author Bo Zhang, Tao Zhang, Yuanyuan Ma, Zesheng Xi, Chuan He, Yunfan Wang was employed by the company State Grid Smart Grid Research Institute Co., Ltd. The remaining authors declare that the research was conducted in the absence of any commercial or financial relationships that could be construed as a potential conflict of interest.

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
