# Peer review of "A Low-Latency Approach for RFF Identification in Open-Set Scenarios"

_electronics, doi:10.3390/electronics13020384_

Round 1
Reviewer 1 Report
Comments and Suggestions for Authors
Explain why the current method (DL-based RFF identification) was selected for the study, and its importance, and compare it with other methods.
The conclusion must discuss the overall importance of the work. Usually, the conclusion section must be self-contained. So some more details are needed conclusion to keep the picture complete.
Reviewer 2 Report
Comments and Suggestions for Authors
Please see the attachment.

Comments on the Quality of English LanguageWith respect to the English language, I consider it is fit for a research journal, even though few errors have been found in the whole manuscript.
Reviewer 3 Report
Comments and Suggestions for Authors
In this paper, the author proposed a scalable RFF identification framework to enhance the learning of representation. In order to preserve the useful information and capture the important features, they construct a feature-exemplar set . Moreover, they adopt a CNN structure to extract the signal from the training dataset. In the experiment part, they build the dataset and compare the performance of different algorithms. The task is interesting and algorithm is quite innovative. After reading this paper, I have following suggestions:
1st In section 5.2, the author propose a contrastive loss between the positive and negative samples. However, they fail to explain it clearly. I suggest the author show us some example to verify the correctness of the proposed loss.
2nd The author introduce each part by introducing the model and method. However, it fails to show the big picture of the task. I suggest the author generate a pipeline plot to show the main purpose of the task.
3rd In section 5.1, the author introduce a multiple layer CNN to extract the signal features. However, it may suffer from gradient diminishing due to the long distance propagation. I suggest the author consider to add attention or borrow the idea from residual network.
4th I suggest the author analyze the robustness toward noise in the experiment part.
5th The writting could be further polished.(e.g in introduction part, the author claim that they can “best characterize the patterns”. It is too strong and arbitrary.)
Comments on the Quality of English Language
Minor revision is required
